# Cognitive Impairment, Chronic Kidney Disease, and 1-Year Mortality in Older Patients Discharged from Acute Care Hospital

**DOI:** 10.3390/jcm9072202

**Published:** 2020-07-12

**Authors:** Mirko Di Rosa, Sonia D’Alia, Francesco Guarasci, Luca Soraci, Elisa Pierpaoli, Federica Lenci, Maddalena Ricci, Graziano Onder, Stefano Volpato, Carmelinda Ruggiero, Antonio Cherubini, Andrea Corsonello, Fabrizia Lattanzio

**Affiliations:** 1Unit of Geriatric Pharmacoepidemiology and Biostatistics, IRCCS INRCA, 60124 Ancona, Italy; m.dirosa@inrca.it (M.D.R.); dalia.soniacs@gmail.com (S.D.A.); guarasci.francesco@gmail.com (F.G.); A.CORSONELLO@inrca.it (A.C.); 2Department of Clinical and Experimental Medicine, University of Messina, 98124 Messina, Italy; 3Advanced Technology Center for Aging Research, Scientific Technological Area, IRCCS INRCA, 60124 Ancona, Italy; e.pierpaoli@inrca.it; 4Unit of Nephrology and Dialysis, IRCCS INRCA, 60124 Ancona, Italy; f.lenci@inrca.it (F.L.); m.ricci@inrca.it (M.R.); 5Department of Cardiovascular and Endocrine-Metabolic Diseases and Aging, Istituto Superiore di Sanità, 00161 Rome, Italy; graziano.onder@iss.it; 6Department of Medical Sciences, University of Ferrara, 44121 Ferrara, Italy; stefano.volpato@unife.it; 7Section of Gerontology and Geriatrics, Department of Medicine, University of Perugia, 06132 Perugia, Italy; carmelinda.ruggiero@unipg.it; 8Geriatria Accettazione Geriatrica e Centro di Ricerca per l’Invecchiamento, IRCCS INRCA, 60124 Ancona, Italy; a.cherubini@inrca.it; 9Unit of Geriatric Medicine, IRCSS INRCA, 60124 Ancona, Italy; 10Scientific Direction, IRCCS INRCA, 60124 Ancona, Italy; f.lattanzio@inrca.it

**Keywords:** Chronic kidney disease, cognitive impairment, constructional praxis, mortality, older

## Abstract

The prognostic interaction between chronic kidney disease (CKD) and cognitive impairment is still to be elucidated. We investigated the potential interaction of overall cognitive impairment or defective constructional praxis and CKD in predicting 1-year mortality among 646 older patients discharged from hospital. The estimated glomerular filtration rate (eGFR) was calculated using the Berlin Initiative Study (BIS) equation. Cognitive impairment was assessed by the Mini Mental State Exam (MMSE) and defective constructional praxis was ascertained by the inherent MMSE item. The study outcome was 1-year mortality. Statistical analysis was carried out using Cox regression. After adjusting for potential confounders, the co-occurrence of eGFR <30 and overall cognitive impairment (Hazard Ratio (HR) = 3.12, 95% Confidence Interval (CI) = 1.26–7.77) and defective constructional praxis (HR = 2.50, 95% CI = 1.08–5.77) were associated with the outcome. No significant prognostic interaction of eGFR < 30 with either overall cognitive impairment (HR = 1.99, 95% CI = 0.38–10.3) or constructional apraxia (HR = 1.68, 95% CI = 0.33–8.50) was detectable, while only cognitive deficits were found significantly associated with the outcome in the interaction models (HR = 3.12, 95% CI = 1.45–6.71 for overall cognitive impairment and HR = 2.16, 95% CI = 1.05–4.45 for constructional apraxia). Overall cognitive impairment and defective constructional praxis may be associated with increased risk of 1-year mortality among older hospitalized patients with severe CKD. However, no significant prognostic interaction between CKD and cognitive impairment could be observed.

## 1. Introduction

Chronic kidney disease (CKD) is known to be associated with relevant morbidity and mortality burden, especially among older people [1]. With increasing age, CKD patients are more likely to develop many comorbidities and CKD-related complications, including cardiovascular and cerebrovascular diseases [2], which can interfere with the management of complex multimorbidity scenarios and increase the risk of poorer quality of life and mortality [2,3].

Furthermore, CKD has formerly emerged as a potential risk factor for cognitive impairment and dementia [4], and several pathways may underly such association. Indeed, diabetes and hypertension (the two most common causes of renal disease) have been shown to be synergistically related to cognitive impairment [5]. Additionally, decreased glomerular filtration rate, polypharmacy, inflammation, anemia, oxidative stress, and renal replacement therapy (RRT) are among the risk factors likely contributing to cognitive dysfunction in CKD [6].

Current data regarding the cross-sectional association between CKD and cognitive impairment are conflicting. Several cross-sectional studies have reported a high prevalence of cognitive impairment among CKD patients compared to those with preserved kidney function [7]. Interestingly, cognitive impairment was found to be cross-sectionally associated with both mild–moderate and severe CKD [8,9]. Low estimated glomerular filtration rate (eGFR) was also associated with a greater prevalence of overall cognitive impairment, constructional praxis deficit, executive dysfunction, and amnestic mild cognitive impairment [10,11]. At variance, no significant association between eGFR and cognitive impairment was observed in a large population of community-dwelling individuals [12]. Anyway, cognitive disorders are currently considered among negative outcomes of CKD [13], and several longitudinal studies confirm this view [7,14,15].

The prognostic impact of CKD has been broadly studied, as it has proven to be a major predictor of mortality among older individuals either in the general population [16,17] or among patients discharged from acute care hospitals [18]. Similarly, cognitive impairment is known to have relevant prognostic implications itself [19,20], but the role of the potential prognostic interaction between CKD and defective cognitive performance has not been fully elucidated. The only study investigating this issue showed that cognitive impairment may increase mortality risk among community-dwelling older people with CKD, but no interaction was found between CKD and low cognitive score [21]. However, an abbreviated Mini Mental State Exam (MMSE) exploring only attention, calculation, and memory [21], which carries a high risk of misclassification bias (i.e., those with the maximum possible score who are identified as cognitively intact individuals might already have cognitive impairment), was used in the former study. Additionally, a recent study of MMSE domains in relation to CKD showed that eGFR may be associated with defective constructional praxis [11], and constructional-executive dysfunction is known to carry negative prognostic implications in selected populations [22,23].

Therefore, we aimed at investigating the potential interaction of overall cognitive impairment or defective constructional praxis and CKD in predicting 1-year mortality among older patients discharged from acute care hospital.

## 2. Materials and Methods

This study uses data from the CRiteria to assess Inappropriate Medication use among Elderly complex patients (CRIME) project, a multicenter prospective observational study carried out in seven geriatric and internal medicine acute care wards throughout Italy [24]. Briefly, all patients consecutively admitted to participating wards between June 2010 and May 2011, were asked to contribute in the study. Exclusion criteria were age <65 years and unwillingness to participate in the study. After obtaining a written informed consent, all participants were assessed within the first 24 h from hospital admission and followed until discharge and data were collected on demographic, socioeconomic, and clinical characteristics, with detailed data collection on pharmacological therapy and comprehensive geriatric assessment. After discharge, patients were reassessed at 3, 6, and 12 months. The study protocol was approved by the Ethics Committee of the Catholic University of Rome (Project identification code: P/582/CE/2009).

Overall, 1123 patients were enrolled in the study. Patients with incomplete baseline data (*n* = 3) and those who died during hospitalization (*n* = 39) were excluded from the present analysis. Patients with incomplete follow-up data (*n* = 274), as well as those with missing Mini Mental State Exam (MMSE; *n* = 137) [25] and serum creatinine (*n* = 30) were also excluded, leaving a final sample of 646 patients to be included in the present analysis.

Patients excluded because of missing data were older (85 ± 10 vs. 80 ± 11, *p* < 0.001) and more frequently affected by dementia (34.8% vs. 13.3%, *p* < 0.001), delirium (13.7% vs. 3.6%, *p* < 0.001) and Basic Activity of Daily Living (BADL) dependency (62.1% vs. 27.1%, *p* < 0.001).

### 2.1. Outcome

1-year mortality was the main outcome of this study. Data on living status during follow-up were obtained by interviewing the patients and/or their formal and/or informal caregivers. Regarding patients who died during the follow-up period, date and place of death were retrieved by relatives or caregivers. The municipal registers were consulted when neither patients or relatives or caregivers could be contacted.

### 2.2. Exposure Variables

Creatinine (Scr) was measured by Isotope Dilution Mass Spectrometry (IDMS) traceable method when the patients were in stable clinical conditions (i.e., usually the day before discharge). eGFR was calculated uding the Berlin Initiative Study (BIS) Equations (1) and (2) [26]:Women: eGFR = (3736 × (Scr) − 0.87 × (age) − 0.95) × 0.82,(1)
Men: eGFR = 3736 × (Scr) − 0.87 × (age) − 0.95,(2)

The BIS equation was chosen for this study because it was specifically developed in a population aged 70 or more [26] and externally validated [27]. Patients were grouped according to kidney function as follows: eGFR >60, 45–59.9, 30–44.9, and <30 mL/min/1.73 m^2^.

Cognitive status was assessed in stable conditions (i.e., the day before discharge) using the 30-item MMSE [25] and overall cognitive impairment was defined as an age- and education-adjusted MMSE total ≤24. Constructional apraxia was defined as scoring 0 at copy of crossed pentagons.

To investigate the impact of cognitive function on the relationship between eGFR and mortality, eGFR at discharge was separately stratified by MMSE score <24 or defective constructional praxis.

### 2.3. Covariates

Age, gender, body mass index (BMI) <18.5 kg/m^2^, serum albumin, depression (Geriatric Depression Scale (GDS) >5) [28], dependency in at least one Basic Activity of Daily Living (BADL) [29], and being bedridden during stay were included in the analysis as potential confounders. Selected diagnoses known to be associated with cognitive impairment and CKD, including diabetes, stroke, dementia, anemia (defined as Hb >12 g/dL for females and Hb <13 g/dL for males) [30], coronary artery disease, heart failure, and chronic obstructive pulmonary disease were also included in the analysis. The number of diagnoses was calculated and considered as an index of over-all comorbidity. The number of medications used during stay was also considered as a potential confounder.

### 2.4. Analytic Approach

First, characteristics of patients according to 1-year mortality were analyzed. Continuous variables were reported as either mean and standard deviation or median and interquartile range on the basis of their distribution assessed by Shapiro–Wilk test. Comparisons between groups was carried out by unpaired Student *t*-test or Mann–Whitney U test, when appropriate. Categorical variables were expressed as number and percentage and significant differences assessed by Chi-square test.

The association between eGFR categories, cognitive impairment (overall or defective constructional praxis), and 1-year mortality was explored by Kaplan–Meyer curves. Therefore, we built multivariable Cox proportional hazard models (crude, age- and gender-adjusted, and fully adjusted for all potential confounders) to obtain an adjusted estimate of the association between exposure variables and study outcome. The fully adjusted analysis was also repeated with a backward stepwise method to verify whether overfitting may affect the multivariable model. In order to account for potential residual confounding, sensitivity analyses were carried out by excluding patients with dementia, stroke or delirium. The interaction terms between cognitive impairment or defective constructional praxis and eGFR categories were investigated by Cox regression analysis. Attrition bias was investigated by age- and gender-adjusted logistic regression analysis of MMSE and eGFR to loss at follow-up. Statistical analysis was carried out using Stata 15.1 Software Package for Windows (Stata Corp, College Station, Lakeway, TX, USA).

## 3. Results

General characteristics of patients grouped according to 1-year mortality are reported in Table 1. Overall, 109 out of 646 patients (16.9%) died during follow-up. Patients who died were older, more frequently bedridden, underweight, and had lower serum albumin levels. Overall burden of comorbidity and polypharmacy was higher among patients who died, as was the prevalence of BADL dependency, dementia, coronary artery disease, heart failure, and anemia. The prevalence of eGFR <30, overall cognitive impairment, and defective constructional praxis was higher among patients who died (Table 1). The Kaplan–Maier curves show a graded reduction of survival in relation to eGFR values, which was much more evident among patients with overall cognitive impairment or defective constructional praxis (Figure 1).

The association between eGFR <30 and mortality in patients with overall cognitive impairment or defective constructional praxis was also confirmed after adjusting for potential confounders (Table 2). Women (Hazard Ratio (HR) = 0.49, 95% Confidence Interval (CI) = 0.32–0.75), age (HR = 1.05, 95% CI = 1.01–1.09), dependency in at least one BADL (HR = 2.13, 95% CI = 1.29–3.54), heart failure (HR = 1.56, 95% CI = 1.00–2.44), and being underweight (HR = 3.22, 95% CI = 1.38–7.48) also qualified as predictor of mortality in the fully adjusted model including overall cognitive impairment and severe CKD. Similarly, women (HR = 0.48, 95% CI = 0.31–0.75), age (HR = 1.05, 95% CI = 1.02–1.09), dependency in at least one BADL (HR = 2.19, 95% CI: 1.32–3.62), and being underweight (HR = 3.40, 95% CI = 1.45–7.98) also qualified as predictor of mortality in the fully adjusted model including constructional apraxia. The association between eGFR <30 and mortality in patients with overall cognitive impairment (HR = 3.15, 95% CI = 1.90–5.22) or defective constructional praxis (HR = 2.76, 95% CI = 1.67–4.55) was confirmed when using backward stepwise method. Other significant predictors of the outcome were substantially unchanged in this latter analysis.

Combined overall cognitive impairment and eGFR <30 were significantly associated with the outcome even after excluding patients with dementia (*n* = 560; HR = 3.40, 95% CI = 1.29–8.99), delirium (*n* = 625; HR = 3.20, 95% CI = 1.29–7.98), or stroke (*n* = 533; HR = 3.32, 95% CI = 1.27–8.70). The corresponding figures for the association of combined defective constructional praxis and eGFR <30 with 1-year mortality were HR = 2.84, 95% CI = 1.10–7.29; HR = 2.47, 95% CI = 1.06–5.74; and HR = 2.88, 95% CI = 1.15–7.24, respectively.

No significant interaction of eGFR <30 with either overall cognitive impairment (HR = 1.99, 95% CI = 0.38–10.3) or constructional apraxia (HR = 1.68, 95% CI = 0.33–8.50) was detectable, while only cognitive deficits were found significantly associated with the outcome in the interaction models (HR = 3.12, 95% CI = 1.45–6.71 for overall cognitive impairment and HR = 2.16, 95% CI = 1.05–4.45 for constructional apraxia). Finally, both MMSE (Odds Ratio (OR) = 0.98, 95% CI = 0.96–1.01) and eGFR (OR = 1.01, 95% CI = 0.99–1.02) were not significantly associated with being lost to follow-up.

## 4. Discussion

The present study shows that the association between eGFR and 1-year mortality is much more evident among patients carrying overall cognitive impairment or defective constructional praxis. Nevertheless, no significant prognostic interaction was observed, and cognitive impairment may have a greater prognostic impact compared to CKD. Our findings add to the present knowledge by showing the relevant prognostic role of cognitive impairment among patients with severe CKD in an unselected population of older patients discharged from hospital and are consistent with previous studies in the general population [21,31,32,33].

Several common potential mechanisms may be involved in the coexistence of cognitive impairment and CKD, as well as their prognostic implications. Vascular atherosclerosis and endothelial dysfunction are commonly present in CKD patients [34] and are involved in the pathogenesis of vascular dementia and stroke [35], which are common causes of cognitive impairment in CKD [36]. Additionally, microvascular renal injury is a prominent feature of diabetes and hypertension, known as common causes of CKD and cognitive impairment [37]. Other mechanisms involve oxidative stress and inflammation, since increased serum levels of free radicals and pro-inflammatory cytokines, including interleukin-6 (IL-6), tumor necrosis factor (TNF), and IL-1beta, contribute to both CKD progression [38,39] and neuroinflammation [40]. Additionally, serum levels of neuropeptide Y (NPY), parathormone (PTH), and fibroblast growing factor 23 (FGF 23) are commonly increased in CKD and seem to promote endothelial dysfunction in brain microcirculation [38]. In contrast, decreased serum concentration of alpha-klotho, the co-receptor for FGF-23, has also been found in CKD [41] and is related to higher risk of dementia and cerebral white matter disease [42]. Comorbidities and complications of CKD, including anemia [43], depression [44], and gastrointestinal dysregulation with gut dysbiosis [45] may contribute to the development of cognitive impairment in CKD. Finally, CKD itself generates a toxic milieu with accumulation of several neurotoxins able to favor the onset of cognitive impairment [38,46].

From the clinical point of view, cognitive impairment may interfere with adherence to CKD risk reduction strategies, including antihypertensive drug use, dietary potassium restriction, compliance with regular follow-up monitoring, and eventual dialysis planning [47,48]. Additionally, cognitive dysfunction may alter the ability to recognize the clinical manifestations of kidney function deterioration, with a consequent increased risk of cardiovascular complications and death [49]. Additionally, the management of selected CKD comorbidities (e.g., diabetes and heart failure) may become very challenging among cognitively impaired patients [50,51], with relevant consequences in terms of mortality risk. Cognitive impairment may worsen adherence to antidiabetic medications [52] and reduce the awareness of hypoglycemic warning symptoms, thus increasing the risk of silent hypoglycemia further contributing to cognitive deterioration [53]. Similarly, cognitive impairment may reduce awareness of disease worsening and adherence to medications among patients with heart failure [54,55], which in turn may increase the risk of hospitalization and death [56]. Our findings, together with the above bulk of evidence, suggests that early identification of cognitively impaired patients with CKD may help to implement specific strategies aimed at preventing further cognitive deterioration and improving long-term management of CKD patients. Despite the potential therapeutic role of physical exercise on cognitive outcomes is more and more recognized, there are only limited data showing that exercise training may provide some potential benefits on cognitive performance among patients with end-stage renal disease in a non-randomized study [57]. Finally, studies on cognitive interventions among CKD patients are still in their stage of infancy [58].

The prognostic role of defective constructional praxis also deserves to be mentioned. Constructional praxis is the complex function organizing cognitive skills into goal-oriented activity. Brain regions involved include the frontal cortex and the right parietal lobes, and lesions of the right insular cortex are known to impair constructional–executive skills [59]. Interestingly, severe CKD was previously found to be associated with decreased cerebral glucose metabolism in several brain areas, including both left and right prefrontal cortex [60], and a significant association between severe CKD and constructional apraxia was observed among older hospitalized patients [11]. Thus, our findings may reflect the involvement of specific brain areas in older patients with severely reduced kidney filtration. Additionally, defective constructional–executive function was found to be associated with an increased risk of death in several different populations, including patients with dementia [61], community-dwelling older individuals [62], and patients with chronic obstructive pulmonary disease [23]. Finally, the insular cortex also contributes to autonomic control, and the right insula is known to be involved in modulating sympathetic function. Right insular lesions are associated with unopposed parasympathetic modulation, leading to increased risk of bradyarrhythmias, prolonged QT intervals, syncope, falls, hip fractures, and death [63], and drawing a impairment (i.e., a test exploring constructional–executive skills) was found to be associated with depressed sympathetic modulation in a former study [64].

Limitations of our study deserve to be acknowledged. First, patients excluded because of missing data were older and more frequently affected by dementia, delirium, and BADL dependency, which may bias study results and reduce their generalizability. Even considering results of sensitivity and potential attrition analyses, we need to recognize that this issue cannot be completely eliminated through extensive statistics. Thus, risk of bias remains a major limitation of the present study. The investigation of the prognostic role of CKD and cognitive impairment was not among the primary aims of the CRIME study. Thus, the definition of CKD and cognitive impairment were obtained a posteriori and adapted for the purpose of the present analysis. Patients’ acute conditions related to hospitalization may have affected estimation of kidney function in our study population. Indeed, even considering that serum creatinine was measured in stable conditions, we could not exclude that even transient changes in hydration status may have affected serum creatinine and creatinine-based eGFR. Selected variables not included in the present study, e.g., the overall quality of post-discharge care and the availability of formal and informal caregivers, may represent a relevant source of residual confounding. MMSE was the only measure of cognitive performance in our study, and we cannot rule out that using an extensive battery of neuropsychologic tests may yield different results. However, MMSE was extensively used for validation studies in the hospital setting [65,66]. Finally, data about the cause of mortality were not collected, which may limit the understanding of differential predictive value of CKD and/or cognitive impairment with regards to specific causes of death (e.g., cardiovascular mortality). Nevertheless, strengths of our study are the inclusion of an unselected population of older hospitalized patients, the assessment of several relevant confounders, and the rigorous analytical method.

## 5. Conclusions

Overall cognitive impairment and defective constructional praxis may be associated with increased risk of 1-year mortality among older hospitalized patients with severe CKD. However, no significant prognostic interaction between CKD and cognitive impairment could be observed. Potential mediators contributing to these findings warrant further investigations. Recognition of cognitive impairment at the time of hospital discharge may provide clinically relevant information, but intervention strategies to better manage cognitively impaired older patients with CKD need to be further investigated.

## Figures and Tables

**Figure 1 jcm-09-02202-f001:**
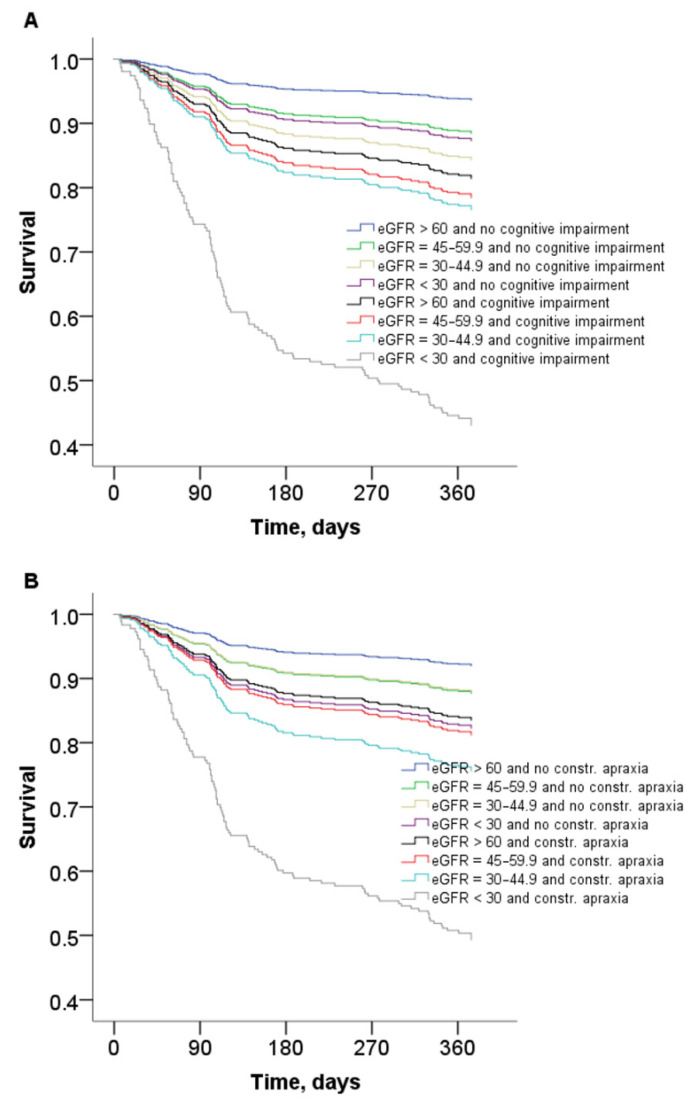
Kaplan–Meier curves showing survival associated with estimated glomerular filtration rate (eGFR) among patients (**A**) with or without overall cognitive impairment (log-rank = 69.9, *p* < 0.001) or (**B**) defective constructional praxis (log-rank = 52.7, *p* < 0.001).

**Table 1 jcm-09-02202-t001:** General characteristics of patients according to 1-year mortality.

	All Patients (*n* = 646)	1-Year Mortality		
		No (*n* = 537)	Yes (*n* = 109)	*p*-Value
Age, years	80 (11)	79 (11)	84 (8)	0.000
Sex, female	52.94	54.56	44.95	0.067
BMI <18.5 kg/m^2^	2.61	1.76	6.73	0.004
Serum albumin, g/dL	3.7 (0.6)	3.7 (0.6)	3.5 (0.8)	0.000
Being bedridden	9.60	6.89	22.94	0.000
Dependency in at least one BADL	27.09	21.60	54.13	0.000
Depression	35.27	34.26	41.25	0.226
Delirium	3.56	3.35	4.59	0.526
Dementia	13.31	12.10	19.27	0.045
Stroke	17.49	16.57	22.02	0.172
Diabetes	30.34	29.42	34.86	0.260
Coronary artery disease	32.51	29.80	45.87	0.001
Heart failure	29.10	25.33	47.71	0.000
Chronic obstructive pulmonary disease	40.09	38.55	47.71	0.075
Anemia	48.92	45.07	67.89	0.000
Number of diagnoses	5 (4)	5 (4)	6 (4)	0.000
Number of drugs	10 (6)	10 (6)	11 (8)	0.039
Glomerular filtration rate BIS, mL/min/1.73 m^2^	56.4 (25.2)	58.3 (24.5)	47.5 (27.7)	0.000
Glomerular filtration rate BIS, mL/min/1.73 m^2^				0.000
60 or more	42.7	45.3	30.3	
45–59.9	31.7	32.2	29.4	
30–44.9	17.7	17.1	20.2	
<30	7.9	5.4	20.2	
MMSE score <24	49.23	44.88	70.64	0.000
Defective constructional praxis	55.88	52.33	73.39	0.000

Data are percentage or median (interquartile range, IQR). *p*-values are referred to Chi-square or Wilcoxon rank-sum test, when appropriate. BMI, body mass index; BADL, Basic Activity of Daily Living; BIS, Berlin Initiative Study; MMSE, Mini Mental State Exam.

**Table 2 jcm-09-02202-t002:** Cox regression analysis of eGFR with or without overall cognitive impairment or defective constructional praxis to 1-year mortality.

		Crude	Age- and Gender-Adjusted	Fully Adjusted *
	Mortality Rate, *n* (%)	HR (95% CI)	HR (95% CI)	HR (95% CI)
Patients without cognitive impairment & eGFR >60	9 (6.2%)			
Patients without cognitive impairment & eGFR 45–59.9	13 (11.3%)	1.86 (0.80–4.36)	1.36 (0.58–3.22)	1.16 (0.48–2.80)
Patients without cognitive impairment & eGFR 30–44.9	8 (15.4%)	2.59 (1.00–6.72)	1.54 (0.58–4.10)	1.26 (0.46–3.42)
Patients without cognitive impairment & eGFR <30	2 (12.5%)	2.06 (0.45–9.54)	1.26 (0.27–6.00)	0.56 (0.10–2.95)
Patients with cognitive impairment & eGFR >60	24 (18.3%)	3.12 (1.45–6.71)	2.56 (1.18–5.56)	1.53 (0.68–3.48)
Patients with cognitive impairment & eGFR 45–59.9	19 (21.1%)	3.68 (1.67–8.14)	2.23 (0.97–5.14)	1.29 (0.53–3.13)
Patients with cognitive impairment & eGFR 30–44.9	14 (22.6%)	4.05 (1.75–9.36)	2.33 (0.97–5.63)	1.08 (0.42–2.77)
Patients with cognitive impairment & eGFR <30	20 (57.1%)	12.79 (5.82–28.11)	7.09 (3.08–16.31)	3.12 (1.26–7.77)
Patients without defective constructional praxis & eGFR >60	11 (7.8%)			
Patients without defective constructional praxis & eGFR 45–59.9	11 (12.1%)	1.58 (0.68–3.64)	1.19 (0.51–2.76)	1.29 (0.55–3.04)
Patients without defective constructional praxis & eGFR 30–44.9	5 (11.9%)	1.55 (0.54–4.47)	0.96 (0.33–2.82)	0.80 (0.27–2.36)
Patients without defective constructional praxis & eGFR <30	2 (18.2%)	2.34 (0.52–10.55)	1.40 (0.30–6.49)	0.61 (0.12–3.20)
Patients with defective constructional praxis & eGFR >60	22 (16.3%)	2.16 (1.05–4.45)	1.84 (0.89–3.83)	1.38 (0.64–2.96)
Patients with defective constructional praxis & eGFR 45–59.9	21 (18.4%)	2.49 (1.20–5.16)	1.54 (0.72–3.28)	1.02 (0.45–2.29)
Patients with defective constructional praxis & eGFR 30–44.9	17 (23.6%)	3.35 (1.57–7.14)	1.88 (0.84–4.17)	1.10 (0.48–2.56)
Patients with defective constructional praxis & eGFR <30	20 (50.0%)	8.46 (4.05–17.66)	4.75 (2.19–10.32)	2.50 (1.08–5.77)

* Adjusted for age, sex, depression, dependency in at least one Basic Activity of Daily Living (BADL), delirium, dementia, stroke, diabetes, coronary artery disease, heart failure, chronic obstructive pulmonary disease, anemia, body mass index (BMI) <18.5 kg/m^2^, being bedridden, number of diagnoses, number of drugs. eGFR, estimated glomerular filtration rate.

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
