# Peer review of "Cognitive Impairment, Chronic Kidney Disease, and 1-Year Mortality in Older Patients Discharged from Acute Care Hospital"

_jcm, 2020, doi:10.3390/jcm9072202_

Round 1
Reviewer 1 Report
Interesting work, observations carried out on a large population patients.
The authors study shows that the association between eGFR and 1-year mortality is much more evident among patients carrying overall cognitive impairment or defective constructional praxis. Authors thought that significant prognostic interaction could be observed and cognitive impairment may have a greater prognostic impact compared to CKD. Overall cognitive impairment and defective constructional praxis may be associated with increased risk of 1-year mortality among older hospitalized patients with severe CKD
Author Response
Interesting work, observations carried out on a large population patients.
The authors study shows that the association between eGFR and 1-year mortality is much more evident among patients carrying overall cognitive impairment or defective constructional praxis. Authors thought that significant prognostic interaction could be observed and cognitive impairment may have a greater prognostic impact compared to CKD. Overall cognitive impairment and defective constructional praxis may be associated with increased risk of 1-year mortality among older hospitalized patients with severe CKD.
AU] We would like to thank the Reviewer for appreciating our work.
Reviewer 2 Report
Thank you for your submission regarding cognitive impairment, chronic kidney disease, and 1-year mortality in older adults. Overall, I found your work to be of quality and well considered. You address an important question but recognize that it is difficult to clearly answer given the many cofounders in the study population and fact that you are using data for which this was not the primary purpose. I especially commend you on your paragraph (pg 8 of 12) regarding the importance of constructional praxis. Like in other chronic conditions (i.e. CHF), I suspect this does play a major role in the patient's ability to successfully manage their CKD.
I would strongly encourage having a native English speaker review your manuscript to improve the flow of the language and grammar.
Author Response
Thank you for your submission regarding cognitive impairment, chronic kidney disease, and 1-year mortality in older adults. Overall, I found your work to be of quality and well considered. You address an important question but recognize that it is difficult to clearly answer given the many cofounders in the study population and fact that you are using data for which this was not the primary purpose. I especially commend you on your paragraph (pg 8 of 12) regarding the importance of constructional praxis. Like in other chronic conditions (i.e. CHF), I suspect this does play a major role in the patient's ability to successfully manage their CKD.
I would strongly encourage having a native English speaker review your manuscript to improve the flow of the language and grammar.
AU] we would like to thank the Reviewer for appreciating our work. Language and grammar have been carefully checked and corrected throughout the manuscript. Additionally, we added a paragraph dealing with the need for future studies about intervention strategies specifically targeting older CKD patients with cognitive impairment (see page 8, lines 236-244).
Reviewer 3 Report
The authors report on the impact of cognitive impairment on 1-year mortality among older hospitalized patients with severe CKD. In their study, no prognostic interaction between CKD and cognitive impairment could be observed.
Comments:
- In the end, this study replicates results from a fromer study showing a negative impact of cognitive impairment on mortality in patients with CKD. This is not a surprising result, given the fact that CKD leads to cognitive impairment and death.
- The introduction is difficult to read because of it's length and complex content. This should be rewritten.
- The huge number of patients excluded from the analysis (nearly half) leads to a bias in the analysis which can't be eliminated through extensive statistics.
- Statistics were very extensive and included a large number of possible confounders, probably leading to an overfitting of the model.
- It is really unclear when kidney function was evaluated and in which condition patients where at time of the blood sample, as authors say, this might lead to a misclassification and underestimation of "real" kidney function.
Author Response
The authors report on the impact of cognitive impairment on 1-year mortality among older hospitalized patients with severe CKD. In their study, no prognostic interaction between CKD and cognitive impairment could be observed.
Comments:
In the end, this study replicates results from a fromer study showing a negative impact of cognitive impairment on mortality in patients with CKD. This is not a surprising result, given the fact that CKD leads to cognitive impairment and death.
AU] We would like to thank the Reviewer for this comment. Besides confirming formerly reported findings, our study adds to current knowledge by showing the primary prognostic role of cognitive impairment compared to CKD. Additionally, our results were obtained by using the full version of MMSE, while the use of abbreviated MMSE only exploring attention, calculation, and memory was the main limitation of the former study. The use of the full version of MMSE also allowed us to explore the prognostic role of defective constructional praxis. Finally, our findings strengthen the need for future studies about intervention strategies specifically targeting older CKD patients with cognitive impairment (see page 8, lines 236-244). We hope that this bulk of new data may be deemed suitable for publication.
The introduction is difficult to read because of it's length and complex content. This should be rewritten.
AU] The Introduction section has been eztensively revised (see page 1 and 2)
The huge number of patients excluded from the analysis (nearly half) leads to a bias in the analysis which can't be eliminated through extensive statistics.
AU] We completely agree with the Reviewer. This limitation was already acknowledged. In the present version we added a sentence to clarify that extensive statistics cannot eliminate potential bias (page 9, lines 275-276)
Statistics were very extensive and included a large number of possible confounders, probably leading to an overfitting of the model.
AU] The fully adjusted model was built considering sevral confounders may affect mortality in a population of older patients discharged from hospital. However, even using a backward stepwise procedure in multivariable analysis, the variables retained in the models were identical to those significantly associated with the outcome when simultaneously including all confounders in the Cox regression models. See page 3, lines 138-139 and page 4, lines 164-168
It is really unclear when kidney function was evaluated and in which condition patients where at time of the blood sample, as authors say, this might lead to a misclassification and underestimation of "real" kidney function.
AU] We clarified that serum creatinine values used to calculate eGFR were measured in stable clinical condition (i.e. usually the day before discharge). See page 3, lines 106-107.
Round 2
Reviewer 3 Report
Dear authors,
the manuscript is now really better than in the first version, thanks for incorporating the comments of the Reviewers.
Author Response
jcm-832803 - Cognitive Impairment, Chronic Kidney Disease and 1-Year Mortality in Older Patients Discharged from Acute Care Hospital – Di Rosa et al
Poin-by-point answers to Editor’s and Reviewer’s criticisms
Editor
The paper definitively improved ,however criticisms still persist especially from one reviewer and the authors should try to sort out and eventually better acknowledge the limitation of the paper in the test
[AU] We would like to thank you for having encouraged us to further revise our paper. Unfortunately, the original design of the CRIME study, as well as the available dataset did not allow to overcome the issue related to risk of bias. For this reason, we changed the order of limitations and emphasized the presistence of risk of bias as the most relevant limitation of our study (lines 276-279).
Reviewer
the manuscript is now really better than in the first version, thanks for incorporating the comments of the Reviewers. The manuscript improved from the first to the second version, but the risk of bias is still there.
[AU] We would like to thank you for having encouraged us to further revise our paper. Unfortunately, the original design of the CRIME study, as well as the available dataset did not allow to overcome the issue related to risk of bias. For this reason, we changed the order of limitations and emphasized the presistence of risk of bias as the most relevant limitation of our study (lines 276-279).
Round 3
Reviewer 3 Report
Thanks for the improvements made to the manuscript